

# Effective protein inhibition in intact mouse oocytes through peptide nanoparticle-mediated antibody transfection

Ruichao Li, Zhen Jin, Leilei Gao, Peng Liu, Zhixia Yang and Dong Zhang

State Key Laboratory of Reproductive Medicine, Nanjing Medical University, Nanjing, Jiangsu, China

## ABSTRACT

Female meiosis is a fundamental area of study in reproductive medicine, and the mouse oocyte model of in vitro maturation (IVM) is most widely used to study female meiosis. To investigate the probable role(s) of an unknown protein in female meiosis, the method traditionally used involves microinjecting a specific antibody into mouse oocytes. Recently, in studies on somatic cells, peptide nanoparticle-mediated antibody transfection has become a popular tool because of its high efficiency, low toxicity, good stability, and strong serum compatibility. However, untill now no researchers have tried using this technique on mouse oocytes because the zona pellucida surrounding the oocyte membrane (vitelline membrane) is usually thought or proved to be a tough barrier to macromolecules such as antibodies and proteins. Therefore, we attempted to introduce an antibody into mouse oocytes using a peptide nanoparticle. Here we show for the first time that with our optimized method, an antibody can be effectively delivered into mouse oocytes and inhibit its target protein with high specificity. We obtained significant results using small GTPase Arl2 as a test subject protein. We propose peptide nanoparticle-mediated antibody transfection to be a superior alternative to antibody microinjection for preliminary functional studies of unknown proteins in mouse oocytes.

# INTRODUCTION

Female meiosis studies in mammals are very relevant to the reproductive health of female humans and have the potential to benefit both basic reproductive medicine and clinical diagnosis and therapies for human reproductive disorders. However, female mammalian meiosis studies are progressing much more slowly than general mammalian mitosis studies (publication in PubMed, 14.1% of general). One of the biggest hurdles is that to knockdown a gene or inhibit a protein, which is usually the first step of the study, researchers have to perform microinjection to introduce siRNA or a specific antibody

Corresponding authors
Zhixia Yang,
yang_zhixia@sina.com
Dong Zhang,
dong.ray.zhang@njmu.edu.cn

into oocytes. Compared with transfection, the disadvantages of microinjection include possible mechanical damage, difficulty with dose control, and a substantially longer time requirement.

Protein inhibition by a specific antibody is one of the most powerful tools in cell biological studies. Gene knockdown by siRNA is very popular in reducing protein level, but it might have off-target since any siRNA might share certain similarity with multiple mRNAs besides the target mRNA. Compared with siRNA, each antigen has its own specific epitope that can be used for the generation of a specific antibody. In addition, protein inhibition can usually take effect much faster than siRNA silencing. Traditionally, microinjecting the corresponding antibody into cells was the only way to inhibit a specific subject protein (*Mehlmann, Jones & Jaffe, 2002*; *Yin et al., 2006*; *Wang et al., 2008*). In recent years, antibody delivery through peptide nanoparticle-mediated transfection has emerged as a superior alternative because of its high efficiency, low toxicity, good stability, and strong serum compatibility (*Kondo et al., 2008*; *Aoshiba, Yokohori & Nagai, 2003*; *Morris et al., 1999*; *Morris et al., 2001*). Another key advantage is that there are diverse commercial antibodies available against a large portion of functionally unknown proteins in the human and mouse proteomes (*Shirai et al., 2014*). However, in oocytes, whether peptide nanoparticle-mediated antibody transfection can effectively deliver antibodies into oocytes has never been tested.

Our lab has been attempting to deliver antibodies into mouse oocytes without the use of microinjection. After screening several peptide nanoparticle transfection reagents and testing many protocols, we have identified the most appropriate one (*Aoshiba, Yokohori & Nagai, 2003*; *Morris et al., 1999*; *Morris et al., 2001*) and successfully developed a feasible standardized approach for use with mouse oocytes.

## MATERIALS AND METHODS

### General chemicals & reagents and animals

Chemicals & reagents were obtained from Sigma unless otherwise stated. ICR mice used in this study were from Vitalriver experimental animal technical co., LTD of Beijing. All animal experiments were approved by the Animal Care and Use Committee of Nanjing Medical University (approval No: 14030158) and were performed in accordance with institutional guidelines.

### Antibodies

Rabbit polyclonal anti-Arl2 (Cat#: 10232-1-AP) and rabbit anti-Arf5 (Cat#: 20227-1-AP) were purchased from Proteintech Inc. (Chicago, IL, USA). Mouse monoclonal anti-α-tubulin antibody (Cat#: sc-8035) antibody was purchased from Santa Cruz Biotechnology, Inc. (Dallas, TX, USA). Human anti-centromere CREST antibody (Cat#: 15-234) was purchased from Antibodies Incorporated (Davis, CA, USA). Cy2-conjugated donkey anti-mouse IgG (Cat#: 715-225-150), Cy2-conjugated donkey anti-rabbit IgG (Cat#: 711-225-152) and Cy3-conjugated donkey anti-human IgG (Cat#: 711-225-152) were purchased from Jackson ImmunoResearch Laboratory (West Grove, PA, USA).

## Oocytes collection and culture

Immature oocytes arrested in prophase I (GV oocytes) were obtained from the ovaries of 3–4 week-old ICR female mice. The mice were euthanatized with $CO_2$ and then sacrificed by cervical dislocation, and ovaries were isolated and placed in operation medium (Hepes) with 2.5 $\mu$M milrinone and 10% fetal bovine serum (FBS; Gibco, Grand Island, NY, USA). Oocytes were released from the ovary by puncturing the follicles with a hypodermic needle. Cumulus cells were washed off the cumulus-oocyte complexes (COC) and every 50 isolated denuded oocytes were placed in 100 $\mu$l droplets of culture medium under mineral oil (Sigma) in plastic dishes (BD). The culture medium was MEM with 0.01 mM EDTA, 0.23 mM Na-pyruvate, 0.2 mM pen/sterep, 3 mg/ml Bovine Serum Albumin (BSA) and 20% FBS (MEM+). Oocytes were cultured at 37.0 °C, 5% $O_2$, 5% $CO_2$ in humidified atmosphere. Prior to IVM (in vitro maturation), all MEM+ include 2.5 $\mu$M milrinone to prevent resumption of meiosis.

## Antibody transfection

For antibody transfection, Chariot™ Protein Delivery Reagent (Active motif, Carlsbad, CA, USA) was used. Briefly, Two tubes, one containing 1 $\mu$l chariot (1 mg/ml in 50% DMSO) in 5 $\mu$l sterile water and the other containing 1 $\mu$g antibody in PBS (final volume is also 6 $\mu$l) were first set up, then solutions from the two tubes were mixed together gently and incubated at room temperature for 30 min to allow the formation of chariot-IgG complex, then the complex solution was added into a 100 $\mu$l MEM+ drop containing 50 oocytes. After 12–14 h treatment, the oocytes was washed to remove the complex-containing MEM+, wait for 1–2 h and in sequence another two rounds of antibody treatment were exerted to ensure the effectiveness of the antibody inhibition. During the whole treatment, typically 40–44 h long, 2.5 $\mu$M milrinone was always included to prevent resumption of meiosis. Next, oocytes were transferred into milrinone-free MEM+ and cultured for 8 or 16 h, then subjected to phenotype analysis-related experiments below. Antibodies for transfection have been thoroughly buffer exchanged (over $10^4$ dilution of original buffer) into PBS/50% glycerol with size-exclusion spin column (cutoff, 100 KDa; spin speed, 5000 rpm; Millipore) to remove antiseptics (usually $NaN_3$) in the original package.

## Immunofluorescence

Oocytes were briefly washed in PBS with 0.05% polyvinylpyrrolidone (PVP), permeated in 0.5% Triton X-100/PHEM (60 mM PIPES, 25 mM Hepes pH 6.9, 10 mM EGTA, 8 mM $MgSO_4$) for 5 min and washed three times rapidly in PBS/PVP. Next the oocytes were fixed in 3.7% paraformaldehyde (PFA)/PHEM for 20 min, washed three times (10 min each) in PBS/PVP and blocked with blocking buffer (1% BSA/PHEM with 100 mM glycine) at room temperature for 1 h. Then the oocytes were in sequence incubated at 4 °C overnight with primary antibody diluted in blocking buffer, washed three times (10 min each) in PBS with 0.05% tween-20 (PBST), incubated at room temperature for 45 min with secondary antibody diluted in blocking buffer (1:750 in all cases), washed three times (10 min each) in PBST. Finally DNA was stained by 10 $\mu$g/ml Hochest 33258

and the oocytes were mounted onto a slide with mounting medium (0.5% propgal gallate, 0.1 M Tris-HCl, pH 7.4, 88% Glycerol) and covered with a cover glass (0.13–0.17 $\mu$m thick). To maintain the dimension of the oocytes, two strips of double-stick tap (90 $\mu$m thick) were sticked between the slide and cover glass. Dilution of primary antibody are as follows: anti-Arl2, 1:200; anti-Arf5, 1:200; anti-$\alpha$-tubulin, 1:500; anti-human centromere, 1:500. The oocytes were examined with an Andor Revolution spinning disk confocal workstation (Oxford instruments, Belfast, Northern Ireland).

### Antibody microinjection

Right before microinjection, antibody buffer exchange was freshly done the same way as for antibody transfection except that the final buffer was PBS with 0.05% PVP (PBS with 50% glycerol keeps antibody more stable but the antibody solution is too sticky to be microinjected). Seven pl of control IgG or Arl2 antibody (1 mg/ml) mixture was injected into each GV oocyte with a IM300 Programmable Microinjector (Narishige, Tokyo, Japan) on Nikon NT-88-V3 Micromanipulation System with heating stage (Nikon, Tokyo, Japan). M2 medium was used to keep stable PH during microinjection. 2.5 $\mu$M milrinone was included to prevent resumption of meiosis.

### Fluorescence intensity measurement

To measure the fluorescence intensity of cytoplasmic rhodamine or spindle microtubules, oocyte immunofluorescence image with rhodamine fluorescence or spindle staining is opened with Image J (National Institutes of Health, Bethesda, MD, USA) and line is drawn around the edge of oocyte cortex or spindle with "polygon selections" tool and the closed line region are measured with "measure" tool. Only the net intensity, which is obtained by subtracting the total intensity with background intensity (Average intensity of the region beside oocyte cytoplasm or spindle), is used for final statistics.

### Data analysis and statistics

All experiments were repeated at least three times, measurement on confocal Images was done with Image J. Data were presented as x ± SEM. Statistical comparison was done with Student's test. $p < 0.05$ was considered to be statistically significant.

## RESULTS

### Peptide nanoparticle-encapsulated antibody can effectively enter intact mouse oocytes

To test whether a peptide nanoparticle-encapsulated antibody can effectively enter intact mouse oocytes, we used a Rhodamine-conjugated control IgG and compared the cytoplasmic fluorescence of control oocytes, oocytes incubated with Rhodamine-IgG only, and oocytes incubated with peptide nanoparticle-complexed Rhodamine-IgG. We treated the GV oocytes and cultured them till MI. As shown in Fig. 1A, the cytoplasmic fluorescence of control oocytes was very low, cytoplasmic fluorescence of oocytes incubated with Rhodamine-IgG only was higher than that of the controls, whereas the cytoplasmic fluorescence of oocytes incubated with peptide nanoparticle-complexed

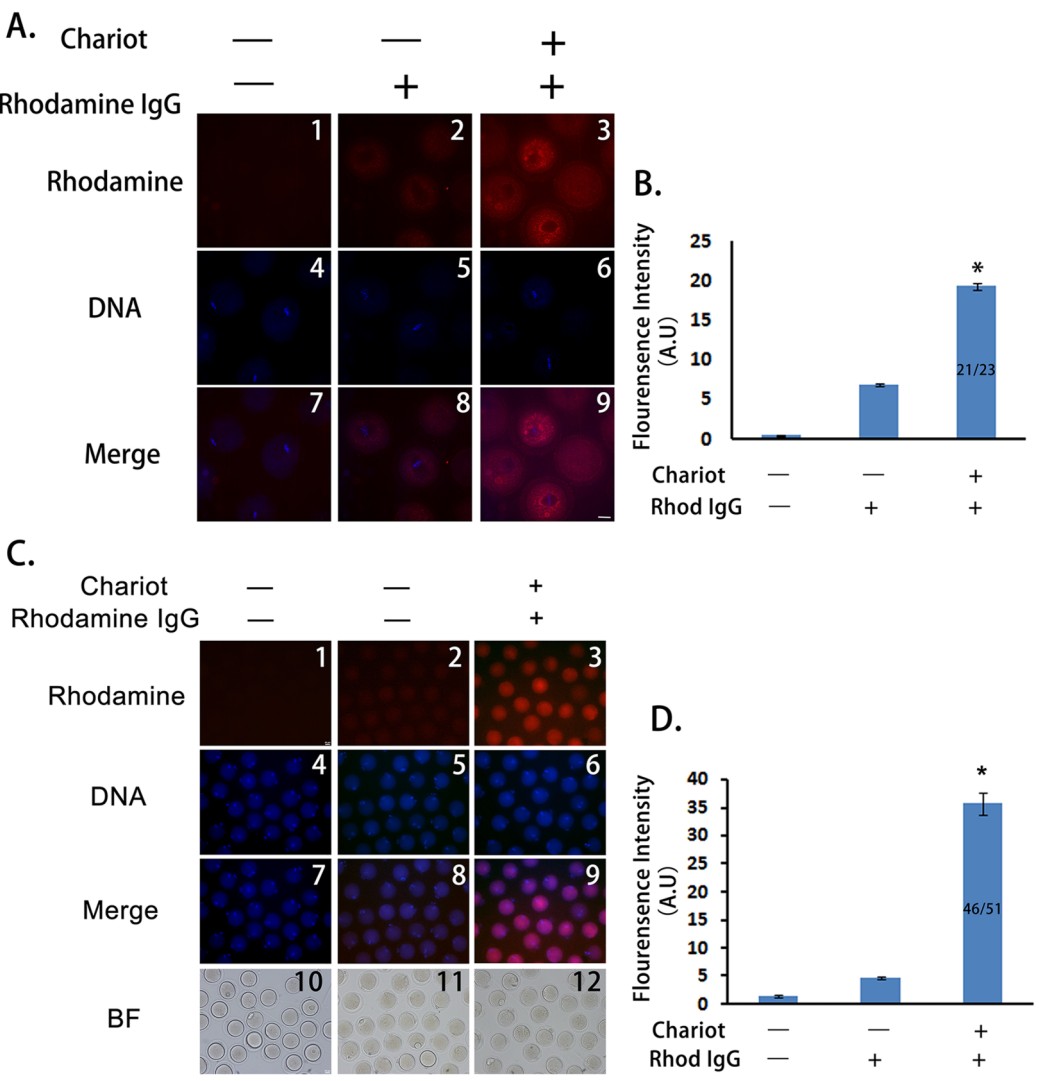

**Figure 1 Peptide nanoparticle-encapsulated antibody can effectively enter intact mouse oocytes.**
(A) Fluorescence comparison between three different groups at MI stage. A1, A4, A7, control oocytes without any treatment; A2, A5, A8, oocytes were incubated with rhodamine-conjugated rabbit IgG; A3, A6, A9, oocytes were incubated with peptide nanoparticle-encapsulated rhodamine-conjugated rabbit IgG; by line, A1–A3, rhodamine-conjugated rabbit IgG in red; A4–A6, DNA in blue; A7–A9, merge of rhodamine and DNA. (B) Quantification of rhodamine fluorescence of three groups in (A). Fraction in the column is number of oocytes with bright rhodamine signal/number of all examined MI oocytes. (C) Fluorescence comparison between three different groups at MII stage. C1, C4, C7, C10, control oocytes without any treatment; C2, C5, C8, C11, oocytes were incubated with rhodamine-conjugated rabbit IgG; C3, C6, C9, C12, oocytes were incubated with peptide nanoparticle-encapsulated rhodamine-conjugated rabbit IgG; by line, C1–C3, rhodamine-conjugated rabbit IgG in red; C4–C6, DNA in blue; C7–C9, merge of rhodamine and DNA. C10–C12, bright field (BF). (D) Quantification of rhodamine fluorescence of three groups in (C). Fraction in the column is the number of oocytes with bright rhodamine signal/number of all examined MII oocytes. Significant comparisons (p < 0.05) marked with asterisks (*). Scale bar, 20 μm.

Rhodamine-IgG was significantly higher than that of the former two groups (Figs. 1A and 1B), indicating that peptide nanoparticle-complexed Rhodamine-IgG was effectively delivered into the oocytes. Over 90% of oocytes (21 of 23) had bright rhodamine signal,

indicating that the transfection efficiency was very good. Furthermore, we had verified that all oocytes from the three different treatments were able to develop to MII stage simultaneously and looked very healthy, and the transfection efficiency was still very good (46 of 51 MII oocytes had bright rhodamine signal), suggesting that the peptide nanoparticle did not harm the mouse oocytes (Figs. 1C and 1D).

## Peptide nanoparticle-mediated antibody transfection can specifically inhibit the target protein

To test whether a peptide nanoparticle-encapsulated antibody can specifically inhibit the target protein while at the same time not affecting others, we selected Arl2 (Arf-like 2) as a target protein and Arf5 (ADP-ribosylation factor 5) as a control protein of the same family. These proteins both belong to the GTP-binding proteins of the Ras superfamily and share high similarity (Fig. 2A). The Arl2 antibody we selected was raised against 2–182 AA of Arl2 (Fig. 2A, blue underlined), whereas the Arf5 antibody was raised against 96–106 AA of Arf5 (Fig. 2A, red underlined). Immunofluorescence showed that both proteins localize along spindle microtubules and share similar localization patterns throughout meiosis (Figs. 2B and 2C). Thus, if Arl2 delivered via peptide nanoparticle became less specific, it could potentially bind to Arf5 as well, in which case Arf5 staining detected by a specific Arf5 antibody would significantly decrease. However, we found no significant difference between the control IgG- and anti-Arl2 antibody-treated groups (Fig. 2D). This result indicates that peptide nanoparticle-mediated antibody transfection can inhibit the target protein with very high specificity, i.e., without affecting other members of the same family.

## Effective protein inhibition through peptide nanoparticle-mediated antibody transfection can be used in protein function analysis

To test whether peptide nanoparticle-mediated antibody inhibition could be a powerful tool in studies of the functions of unknown proteins, we analyzed the meiotic phenotype after the antibody inhibition of Arl2. Because Arl2 mainly localizes within spindles, we hypothesized that it may function in organizing spindles so that the loss of function could affect spindle integrity and meiosis. Thus, we did a systematic phenotypic analysis on the meiotic spindles at 8 and 16 h of IVM. At 8 h, there were significantly more oocytes with clumped chromosomes and without discernable chromatids and spindle microtubules in the Arl2 inhibition group than in the control group (Figs. 3A and 3B, control vs. Arl2, 0.95% vs. 50.18%). Since these chromosomes were distinct from GV chromatins but had no apparent microtubules, we called these oocytes as "GV-like" oocytes. There were also significantly fewer oocytes at MI (Fig. 3B, control vs. Arl2, 44.31% vs. 3.07%) in the Arl2 inhibition group than in the control group. Moreover, the spindle length of MI oocytes in the Arl2 inhibition group was significantly shorter than that in the control group (Figs. 3C and 3D, control vs. Arl2, 12.30 μm vs. 9.43 μm). At 16 h, there were significantly fewer MII (oocytes with first polar body) oocytes (Figs. 3E and 3F, control vs. Arl2, 53.33% vs. 26.19%) in the Arl2 inhibition group than in the control group. Furthermore, significantly more MII oocytes in the Arl2 inhibition group

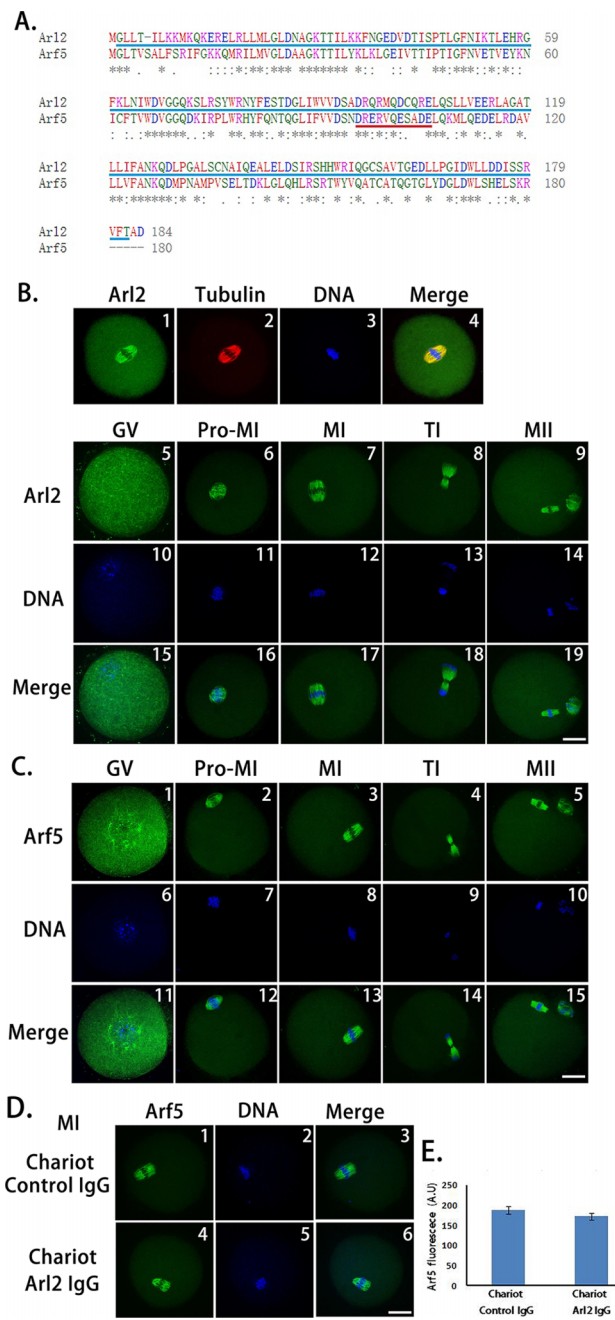

**Figure 2 Peptide nanoparticle-mediated antibody transfection can specifically inhibit target protein in mouse oocytes.** (A) Protein sequence alignment of Arl2 and Arf5. Blue-underlined Arl2 sequence (2–182 AA) are antigen regions for the anti-Arl2 antibody, red-underlined Arf5 sequence (96–106 AA) are antigen regions for the anti-Arf5 antibody. B1–B4, co-localization of Arl2 and microtubules; B5–B19, Arl2 immunolocalization in mouse oocytes at each meiotic stage. Respectively, by column, B5, B10, B15, at GV stage; B6, B11, B16, Pro-MI; B7, B12, B17, MI; B8, B13, B18, TI; B9, B14, B19, MII; by line, B5–B9, Arl2 in green; B10–B14, DNA in blue; B15–B19, merge of Arl2 and DNA. (C) Arf5 immunolocalization in mouse oocytes at each meiotic stage. C1, C6, C11, at GV stage; C2, C7, C12, Pro-MI; C3, C8, C13, MI; C4, C9, C14, TI; C5, C10, C15, MII; by line, C1–C5, Arl2 in green; C6–C10, DNA in blue; C11–C15, merge of Arf5 and DNA. (D) Immunostaining of Arf5 at MI stage in control IgG (D1–D3) or anti-Arl2 antibody (D4–D6) transfection group. (E) Quantification of Arf5 fluorescence at MI stage in control IgG (left) or anti-Arl2 antibody (right) transfection oocytes. Scale bar, 20 μm.

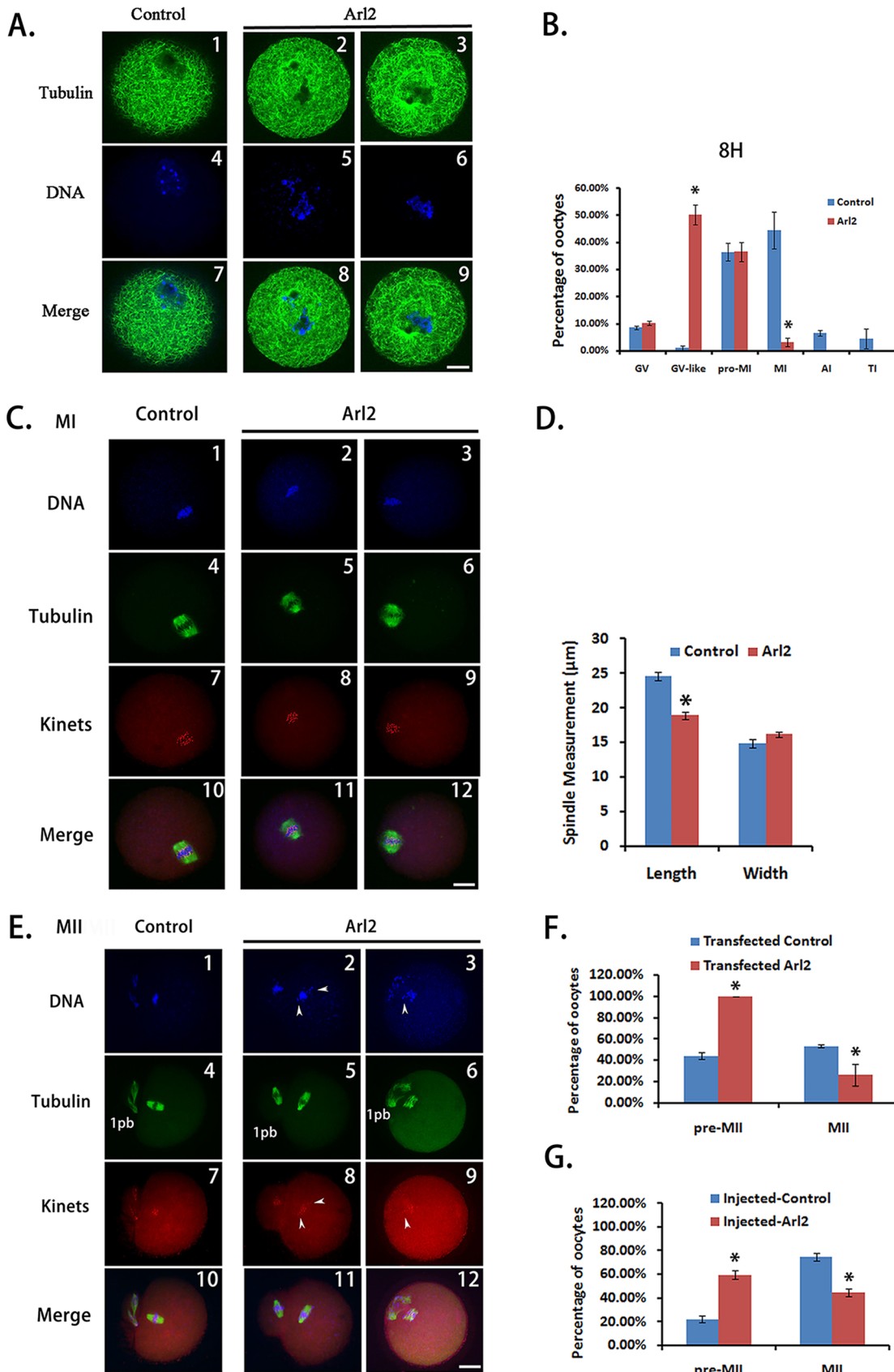

**Figure 3 Effective protein inhibition through peptide nanoparticle-mediated antibody transfection can be used in protein function analysis in mouse oocytes.** (A) At the 8th h of in-vitro maturation (IVM), there were significantly more oocytes with clumped chromosomes and without discernable chromatids and spindle microtubules in the Arl2 inhibition group than in the control group. Since these chromosomes were distinct from GV chromatins but had no apparent microtubules, we called these oocytes "GV-like" oocytes. A1, A4, A7, control group; A2, A5, A8 and A3, A6, A9, Arl2 inhibition group; by line, A1–A3, tubulin in green; A4–A6, DNA in blue; A7–A9, merge of tubulin and DNA. (B) Percentage of oocytes at different stages at the 8th h of IVM. (C) MI spindle in anti-Arl2 antibody transfection group was significantly shorter than in control. C1, C4, C7, C10, control group; C2, C5, C8, C11 and C3, C6, C9, C12, Arl2 inhibition group; by line, C1–C3, DNA in blue; C4–C6, tubulin in green; C7–C9, kinetochores (kinets) in red; C10–C12, merge of DNA, tubulin and kinetochores. (D) Quantification of MI spindle length and width in control or anti-Arl2 antibody transfection group. (E) At the 16th h of IVM, the percentage of MII (1PB) oocytes in anti-Arl2 antibody transfection group was significantly lower than in control. And significantly more MII oocytes in the Arl2 inhibition group had uncongressed chromosomes and kinetochores (arrow pointed). Since these oocytes had finished polar body extrusion but the chromosomes hadn't been well aligned, we called these oocytes "premature MII oocytes" and simplified to "pre-MII oocytes." E1, E4, E7, E10, control group; E2, E5, E8, E11 and E3, E6, E9, E12, Arl2 inhibition group; by line, E1–E3, DNA in blue; E4–E6, tubulin in green; E7–E9, kinetochores (kinets) in red; E10–E12, merge of DNA, tubulin and kinetochores. (F) Percentage of MII (1PB) or Pre-MII oocytes at 16 h of IVM after control IgG or anti-Arl2 antibody transfection. (G) At 16 h of IVM after control IgG or anti-Arl2 antibody, the percentage of MII (1PB) oocytes in anti-Arl2 antibody microinjection group was significantly lower than in control IgG group. There were significantly more pre-MII oocytes in anti-Arl2 antibody microinjection group than in control. Pre-MII oocytes are oocytes with unaligned chromosomes; normal MII oocytes are oocytes with well-aligned chromosomes; MII (1PB) oocytes are all oocytes with first polar body, number of MII (1PB) oocytes = number of Pre-MII oocytes + number of normal MII oocytes. Significant comparisons ($p < 0.05$) were marked with asterisks (*). Scale bar, 20 μm.

had uncongressed chromosomes and kinetochores (Fig. 3E, arrow pointed). Since these oocytes had finished polar body extrusion but the chromosomes hadn't been well aligned, we called these oocytes as "premature MII oocytes" and simplified as "pre-MII oocytes" (Fig. 3F, control vs. Arl2, 44.15% vs. 100%). To further confirm the phenotype, we also microinjected Arl2 antibody into GV oocytes and examined the percentage of oocytes at 16 h (Fig. 3G). Although the absolute values by microinjection (Percentage of MII oocytes, control vs. Arl2, 74.90% vs. 44.75%; percentage of pre-MII oocytes, control vs. Arl2, 22.19% vs. 59.39%) are different from by transfection, the tendency is very similar (Fig. 3G). These results indicate that Arl2 does function in organizing spindles, thereby promoting meiosis progression.

## DISCUSSION

To our knowledge, this is the first study showing that through peptide nanoparticle-mediated antibody transfection, antibodies can effectively enter intact oocytes and inhibit specific proteins without affecting other members of the same family. The Chariot peptide used here is a short amphipathic peptide carrier, it originates from a fusion protein including protein transduction domain (PTD) and TAT protein from human immunodeficiency virus (HIV-1) but been significantly improved so that it can form non-covalent bonds with the antibody, stabilizes the antibody, protects it from degradation, and preserves its natural characteristics during the transfection. The delivery is independent of the endosomal pathway, which can modify macromolecules during internalization. Therefore, chariot delivery can fully keep the antibody at its original form. After delivery, the Chariot peptide-antibody complex dissociates and releases the antibody (Aoshiba, Yokohori & Nagai, 2003; Morris et al., 1999; Morris et al., 2001). We have also successfully established a standardized protocol for the effective delivery of siRNA into mouse oocytes through peptide nanoparticle-mediated siRNA transfection. However,

the maximum reduction percentage of a target protein by siRNA is usually only approximately 70%, and if a protein is very abundant, the remaining protein can still be enough to retain its normal function. Therefore, the optimal approach for the knockdown of a highly rich protein is a combination of siRNA knockdown and antibody inhibition. If the protein content is low, antibody inhibition alone may be enough. In summary, nanoparticle-mediated antibody inhibition is an effective approach to use when studying an unknown protein in mouse oocytes.

Arl2 belongs to the ARF family of small GTP-binding proteins of the Ras superfamily, it interacts with the tubulin-specific chaperone protein known as cofactor D and is involved in the folding of tubulin peptides (*Shern et al., 2003*). In mitosis, Arl2 is present in centrosomes and regulates tubulin polymerization, thereby affecting cell cycle progression (*Zhou et al., 2006*; *Beghin et al., 2007*). However, its role in female meiosis has never been addressed. In the current study, we found that Arl2 localized within spindles and that the inhibition of Arl2 caused substantial spindle defects and significantly delayed meiosis progression. This demonstrates that peptide nanoparticle-mediated antibody inhibition can be used in meiosis studies in mouse oocytes.

In conclusion, for the first time, we have successfully developed peptide nanoparticle-mediated antibody transfection for effective protein inhibition in mouse oocytes. We believe that this new tool will promote further meiosis studies in mouse oocytes.

## ACKNOWLEDGEMENTS

We thank Professor Youqiang Su (State key lab of reproductive medicine, Nanjing Medical University) for discussion and advice about this project.

### Funding

This research is supported by the National Key Basic Research Program of China ("973" program, from Ministry of Science and Technology of China, Grant No: 2013CB945504), the General Program of National Natural Science Foundation of China (from National Natural Science Foundation Committee of China, Grant Nos: 31271441 and 31471406). The funders had no role in study design, data collection and analysis, decision to publish, or preparation of the manuscript.

### Grant Disclosures

The following grant information was disclosed by the authors:
Ministry of Science and Technology of China: 2013CB945504.
National Natural Science Foundation Committee of China: 31271441 and 31471406.

### Competing Interests

The authors declare that they have no competing interests.

# PeerJ

## Author Contributions

- Ruichao Li performed the experiments, analyzed the data, wrote the paper, prepared figures and/or tables, reviewed drafts of the paper.
- Zhen Jin performed the experiments, reviewed drafts of the paper.
- Leilei Gao performed the experiments, reviewed drafts of the paper.
- Peng Liu performed the experiments, reviewed drafts of the paper.
- Zhixia Yang conceived and designed the experiments, contributed reagents/materials/analysis tools, reviewed drafts of the paper.
- Dong Zhang conceived and designed the experiments, contributed reagents/materials/analysis tools, wrote the paper, prepared figures and/or tables, reviewed drafts of the paper.

## Animal Ethics

The following information was supplied relating to ethical approvals (i.e., approving body and any reference numbers):

Animal Care and Use Committee of Nanjing Medical University (approval No: 14030158).

## Data Deposition

The raw data has been supplied as Supplemental Dataset Files.

## Supplemental Information

Supplemental information for this article can be found online at http://dx.doi.org/10.7717/peerj.1849#supplemental-information.

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
