# Peer review of "Effective protein inhibition in intact mouse oocytes through peptide nanoparticle-mediated antibody transfection"

_PeerJ, doi:10.7717/peerj.1849_

## Round 0.1 · original submission · Minor Revisions

Your mansucript was evaluated by 4 reviewers, three give positive reviews, but one gives negative review. I read all reviews carefully, and found that although the last reviewer raised 5 major concerns, I believe it is not difficult to address. I thus recommend you to revise the manuscript taking all 4 reviewers' comments.

Reviewer 1 ·

Basic reporting

No Comments

Experimental design

No Comments

Validity of the findings

No Comments

Additional comments

Manuscript by Li et al attempted to apply a newly developed method "peptide nanoparticle-mediated antibody transfection" in somatic cells to the meiotic research in mammalian oocytes. They verified that antibodies could be effectively delivered into the mouse oocytes to specifically inhibit the target proteins, but had little effect on other proteins including those with highly similar sequences. This optimized method exhibits high efficiency, low toxicity and good stability compared to the conventional way of introducing the antibodies to the oocytes such as microinjection. The experiment design in the present manuscript is sound and the data are convincing. Thus I recommend it for publication in Peer J following the minor revision with English grammar errors and typos.

Reviewer 2 ·

Basic reporting

No comments

Experimental design

No comments

Validity of the findings

No comments

Additional comments

This manuscript aims to an important question on how to develop a rapid, economical and efficient technique to study the cellular function(s) of specific protein in mammalian oocytes. The authors introduced antibody into mouse oocytes using a peptide nanoparticle, and specifically inhibited the antibody-targetting protein, resulting in specific phenotype. Overall, the peptide nanoparticle-mediated delivery of antibody is a simple and available alternative to traditional microinjection approach. The study is well designed and technically sound, the results are clearly presented and support the conclusion.
Additional minor suggestions that I have:
1. In this study, the authors sacrificed mice by cervical dislocation, is this method approved by their institute? Currently, laboratory mice are suggested to be euthanatized with CO2.
2. The delivery of antibody into oocytes by microinjecting is widely used and efficient method for functional analysis of protein in oocytes, I wonder if the authors conducted a parallel experiment of microinjecting Arl2 antibody and compared the specificity and efficiency of phenotype with that induced by peptide nanoparticle antibody transfection, if possible, this kind of data will be a great support.
3. The authors mentioned that the chariot peptide-antibody complex can dissociates and releases the antibody (Line170-171), it is necessary to present some information to explain the mechanism, that will be much helpful for understanding.
4. Milrinone is used to arrest mouse oocytes at prophase stage of meiosis, and in my experience, the working concentration of this agent is usually around 2.5 microM, however, 2.5 nM milrinone was applied in this manuscript, is it true?
5. FBS (Line 71), MEM+ (Line 75), BSA (Line 71) should be written in there full name when they were first introduced.
6. Line 71, “Isolated” should be written as “isolated”.
7. Line 96, in “MgSO4”, "4" should be written in subscript.
8. Line 105, Tris-Hcl” should be written as “Tris-HCl”, and “PH7.4” should be “pH7.4”.
9. Line 112, “Measurement” should be written as “measurement”.
10. Line 113, the manufacturer and its address of Image J should be define.
11. Line 159, What’s the meaning of “….fewer total MII oocytes….”? “total” should be deleted from this text.
12. Line 168, something must be wrong in the spelling of “aChariot peptides”.
13. Line 186, I feel difficult to understand “….Arl2 localized within spindles……”, please rewrite this sentence.
14. Figure 1, in legend “p□ 0.05” must be “P<0.05”; the individual images should be numbered with letters in lowercase, this also should be applied in images in Figure 2 and Figure 3; the fluorescence intensity of DNA is not consistent among three groups; Data of the difference in intracellular Rhodamine signal between groups was presented, how about the number of individual oocytes with bright signal of Rhodamine? Did all the oocytes exhibit positive red signal in nanoparticle transfection group?
15. Figure 2 B,C,D and Figure 3, the images were not merged.
16. The aothors need to define the concepts of “GV-like” and “Pre-MII” in main text and figure legends.

Reviewer 3 ·

Basic reporting

No Comments.

Experimental design

No Comments.

Validity of the findings

No Comments.

Additional comments

The manuscript by Li et al., ”Effective protein inhibition in intact mouse oocytes
through peptide nanoparticle-mediated antibody transfection” tries to establish an efficient method-peptide nanoparticle-mediated antibody transfection to study functions of protein in mouse oocytes. The authors found that antibodies could be effectively delivered into mouse oocytes based their optimized method and inhibited its target protein with high specificity. The manuscript may provide a useful method to study the functions of protein in mouse oocytes with high efficiency.
Minor comments:
1. For Figure 3, how to define the “GV-like”, ”pre-MI”, ”pre-MII” oocytes repectively?
2. For antibody transfection experiment, are the antibodies purified to exclude the potential toxic chemicals in commercial antibodies? Is it possible that the potential toxic chemicals could affect the transfection efficiency or co-transfected into oocytes?

Reviewer 4 ·

Basic reporting

No Comments

Experimental design

No Comments

Validity of the findings

No Comments

Additional comments

Comments to the Authors
The study by Li, et al. entitled "Effective protein inhibition in mouse oocytes through peptide nanoparticle-mediated antibody transfection". Here, the authors have tested Chariot™ Protein Delivery Reagent, in its capacity to transfer antibody into mouse oocytes. The final endpoint was Chariot™ Protein Delivery Reagent can be effective delivered antibody into oocyte and inhibit its target protein with high specificity. The paper has the following weaknesses:
Major comment:
1) The authors showed that peptide nanoparticle-mediated antibody transfection to be a superior alternative to antibody microinjection for preliminary functional studies of unknown proteins in mouse oocytes. Did authors compare other commercial protein transfection kit? How about the efficiency?
2) Line 126: The authors found that all oocytes from the three groups looked very healthy and developed to MII stage simultaneously. Please provide data and make discussions.
3) Line 142: “This results indicated that peptide nanoparticle-mediated antibody transfection can inhibit the target protein with very high specificity.” However, there is no data illustrate the results.
4) Fig.1 and Fig.2: It is not really clear how fluorescence intensively was quantified.
5) Fig.3A: In GV stage, α-tubulin located in nuclear region?
Please consult “Kim B et al. The role of MATER in endoplasmic reticulum distribution and calcium homeostasis in mouse oocytes [J]. Developmental biology, 2014, 386(2): 331-339”.

Minor comment:
1) Line 35: “Compared with siRNA-mediated gene silencing, inhibition by an antibody is more specific ……” Please provide references.
2) Line 48: The authors showed that they have tested many protocols and identified the most appropriate one. However, there are no data or literatures shows that efficiency of peptide nanoparticle transfection reagents in oocytes.
3) The authors found that Arl2 localized within spindles. I suggest that you provide Arl2 and spindle co-localization figure or references in discussion sections
4) Fig.1 is not clearly shown.

---

## Round 0.2 · accepted · Accept

The manuscript has been much improved, and I am delighted to accept it.